# Functional Upper Airway Space Endoscopy: A Prognostic Indicator in Obstructive Sleep Apnea Treatment with Mandibular Advancement Devices

**DOI:** 10.3390/ijerph18052393

**Published:** 2021-03-01

**Authors:** Giulio Gasparini, Gianmarco Saponaro, Mattia Todaro, Gabriele Ciasca, Lorenzo Cigni, Piero Doneddu, Camillo Azzuni, Enrico Foresta, Paolo De Angelis, Giorgio Barbera, Roberta Gaia Parcianello, Horia Vasile Hreniuc, Alessandro Moro

**Affiliations:** 1Maxillofacial Surgery Unit, Faculty of Medicine, University Hospital ‘A. Gemelli’, Catholic University of the Sacred Heart, IRCCS, 00168 Rome, Italy; giulio.gasparini@policlinicogemelli.it (G.G.); gianmarco.saponaro@gmail.com (G.S.); piero.doneddu@gmail.com (P.D.); dottazzuni@virgilio.it (C.A.); enrico.foresta@policlinicogemelli.it (E.F.); Giorgiobarbera87@gmail.com (G.B.); robertagaiaparcianello@gmail.com (R.G.P.); mrolsn@libero.it (A.M.); 2Dipartimento di Neuroscienze, Sezione di Fisica, University Hospital ‘A. Gemelli’, Catholic University of the Sacred Heart, IRCCS, 00168 Rome, Italy; gabriele.ciasca@unicatt.it; 3Department of Odonto-Stomatology, Azienda Ospedaliera Valtellina e Valchiavenna, 23100 Sondrio, Italy; lorenzocigni1@virgilio.it; 4Departement of Head, Division of Oral Surgery and Implantology, Neck and Sensory Organs, University Hospital ‘A. Gemelli’, Catholic University of the Sacred Heart, IRCCS, 00168 Rome, Italy; dr.paolodeangelis@gmail.com; 5Department of Anesthesia and Intensive Care, Fondazione Policlinico Universitario Agostino Gemelli IRCCS, 00168 Roma, Italy; horiavasile.hreniuc@policlinicogemelli.it

**Keywords:** OSA, OSAS, MAD, mandibular advancement, AHI, upper airways

## Abstract

Purpose: The use of a mandibular advancement device (MAD) in the treatment of obstructive sleep apnea (OSA) is a consolidated therapy. This study aimed to evaluate the predictive value of awake upper airways (UA) functional endoscopy in identifying the outcome of MAD therapy. Methods: This observational prospective study included 30 adult OSA patients, all patients underwent pre-treatment awake UA functional endoscopy, during the exam subjects were instructed to advance their mandible maximally, and they were divided into three different groups according to the response of the soft tissue, group A (expansion), group B (stretch), group C (unchanged). The results of this test were used in combination with other noninvasive indexes to predict the treatment outcome in terms of apnea-hypopnea index (AHI) reduction. Results: We found that a substantial AHI reduction occurred in group A and group B while e slight AHI reduction was measured in group C. Conclusion: Based on our experience the awake UA endoscopy is a valid prognostic exam for discriminating responder and non-responder patients; in addition our results indicate the possibility of predicting a range of post-treatment AHI index values.

## 1. Introduction

Several treatment options are available for the effective management of obstructive sleep apnea (OSA). Surgery plays an important role and various authors have described many techniques with good results [1,2,3,4,5]. In addition to surgical therapy, there are other therapeutic options aimed at improving the symptoms and quality of life of patients, among these the use of mandibular advancement devices (MAD) in the treatment of obstructive sleep apnea (OSA) is a consolidated therapy [6]. MADs are indicated in cases of mild and moderate OSA, while they remain one of the few therapeutic options in patients with severe OSA who are unable to undergo continuous positive airway pressure (cPAP) therapy and refuse surgical treatment. The validity of MAD treatment is now supported by a large international group of studies [7], but there are many uncertainties in discriminating responders from non-responder patients.

Many authors [8,9] consider drug-induced sleep endoscopy (DISE) an optimal diagnostic test for evaluating the validity of MAD therapy. DISE allows direct visualization of obstruction sites and collapse patterns in patients affected by snoring or OSA and can be useful to discriminate those who would benefit from MAD therapy. The technique requires a light forward maneuver of the mandible (pull up) to identify improvements in the upper airways (UA) space and reduction of snoring and OSA index. Patients with a sufficient increase in UA width and a reduction in snoring may be valid candidates for MAD therapy [8]. However, the use of DISE in the therapeutic approach to MAD treatment has limitations linked to the need for hospitalization, the risks associated with the use of anesthesiology drugs, and costs. Although the DISE remains the optimum at the moment in the indication for the oral appliance, the use of a less invasive, easy to use, and inexpensive method is necessary. As an alternative to DISE, in our clinical practice, we performed awake UA functional endoscopy and evaluated the response of soft tissue during forwarding positioning of the mandible. We performed an observational study in patients with OSA to validate the predictive role of awake functional endoscopy in MAD therapy.

## 2. Material and Methods

### 2.1. Patients

This observational prospective study included adult OSA patients that were referred to the UOC MaxilloFacial Surgery, IRCCS Foundation Policlinico A. Gemelli of Rome-Italy, from June 2017 to December 2019.

The study was conducted on 30 patients, including 16 males and 14 females, with a mean age of 52.93 ± 8.26 years.

### 2.2. Inclusion and Exclusion Criteria

The study included adult patients with OSA symptoms who had an apnea-hypopnea index (AHI) > 5 (on PSG).

All patients were CPAP intolerant and refused surgical any treatment.

Patients on CPAP treatment, patients undergoing diet therapy for weight loss, or that had previously undergone surgical treatment for OSA were excluded.

All patients were aged >18 years.

All patients had an indication for MAD therapy after evaluation by a dentist expert in sleep medicine.

### 2.3. Patients Evaluation

After recording the medical history, all patients underwent a general, dentist and upper airway examination, Epworth Sleepiness Scale (ESS) analysis, and Obstructive Airway Adult Test (OAAT); the OAAT questionnaire has shown to be a valid tool for the diagnosis of mild (OAAT score ≥ 38; AHI ≥ 5), moderate OSA (OAAT score ≥ 57; AHI ≥ 15), and severe OSA (OAAT score ≥ 73; AHI ≥ 30) [10]. These evaluations were repeated six months post-treatment, (Table 1). The oral examination included a detailed assessment of the dentition, soft palate and uvula, and the size of the tonsils, with a specific estimation of the size of the base of the tongue. Mandibular movements were evaluated in the forward and lateral projections.

### 2.4. Endoscopy and Evaluation Methods

Endoscopy was performed using a flexible nasoendoscope (FN) (ENF-V2, 3.2 mm diameter; Olympus, Germany). The computerized system included a camera and a light source. The software enabled video recording.

The patients were invited to lie down on the ambulatory bed. After the lubrication of the FN, we inserted it into the nasal cavity and proceed with the exploration of the UA. In O space (oropharynx), we invited the patient to simulate snoring. This maneuver gave us an indication of what happens at level O during snoring. Furthermore, this maneuver allowed the patient to relax, accustom himself to the instrument’s presence and stay less contracted.

We invited the patient to advance the mandible maximally and we observed the soft tissue response. The soft tissues of level O can respond to advancement in the following ways:-with a significant (>25%) widening of the airspace [enlargement-expansion (e)];-with a stretch (<25%) of tissue only slightly affecting the airspace [stretch (s)];-without a response, i.e., the airway’s volume remains unchanged [unchanged (U)].

We performed the Muller maneuver to evaluate the lateral collapsibility of the UA. We stopped for a few seconds at level H (hypopharynx, base of the tongue) to allow the patient to relax and get ready for the descent of our instrument. At level H, we evaluated the position of the lingual base towards the epiglottis and both structures towards the posterior wall of the pharynx. We invited the patient to advance the mandible maximally and observed what happened at the base of the tongue and epiglottis. The soft tissues of level H can respond to advancement in the same way as those of the O space. During the examination, we evaluated any signs of gastroesophageal reflux disease (GERD) and evaluated the presence of any neoformations to report to an ENT colleague for further diagnostic investigations. Returning to the O level, the patient was again invited to advance the mandible maximally, and we observed the soft tissue response. We determined whether the response of the soft tissues to mandibular advancement was improved or remained unchanged. After removing the flexible endoscope from the explored nostril, we introduced it into the contralateral nostril and performed the examination only at level N. To evaluate the obstruction of the UAs, we have applied the NOHL classification presented by Vicini et al. [11] in 2012. At levels O and H, we recorded “e”, “s”, or “u” to report the response of the soft tissue to mandibular advancement.

### 2.5. Polysomnography

Patients underwent standard full-night PSG (electron-encephalogram, oculography eye-tracking, electromyogram, oronasal flow, pulse oximetry, respiratory effort, position, electrocardiogram, snoring). The obtained tracks were interpreted by a qualified sleep disorder technician using the diagnostic criteria established by the AASM in 2007 and then revised in 2012 [12,13]. The PSG was performed pre-treatment and six months post-treatment.

### 2.6. Oral Device

A two-piece appliance used to move the mandible forward was custom-made for each patient. Mandibular advancement was set at 60% of the maximum mandibular protrusion at first. The patient was then re-evaluated every 30 days and the appliance was incrementally titrated to the level of protrusion that allows the best improvement of snoring and daytime symptoms without causing discomfort. After 6 months PSG was repeated. If the patient reported no improvement of snoring and daytime symptoms, the MAD advancement was increased by 1 mm, and the patients were re-evaluated every 15 days, if at 90% of the maximum protrusion the patient still reported no improvement, a confirmation PSG was performed and therapy was suspended.

### 2.7. Statistical Analysis

Statistical analyses were performed with the software R (The R Foundation for Statistical Computing, Vienna, Austria) (3.5.2 release) [14]. The R package ggplot2 was used for data visualization. Several markers were considered to evaluate the effectiveness of the MAD. These included the AHI, Epworth scale, snoring index, and OAAT score. Each marker was measured before and after treatment. Age and sex were also taken into account. The selected markers were tested for normality by a visual inspection of the QQ-plot followed by a Shapiro–Wilk test (data not shown). This analysis showed that the normality assumptions were not always met. Accordingly, variables were reported as the median (Q2) and interquartile range (IQR), and nonparametric tests were used to compare different groups. Calculations were performed with the R function stat_compare_means. The presence of statistically significant differences between nonindependent datasets was assessed with the paired-sample Wilcoxon test. Comparisons between multiple independent samples were performed using the Kruskal–Wallis test followed by a post hoc analysis with the Wilcoxon unpaired two-sample test. In the case of heteroscedastic data, as assessed with the Levene Test, these comparisons were also performed with Welch’s ANOVA followed by a Games–Howell post hoc test for multiple means comparisons (data not shown). In each of the investigated cases, the two analyses allowed us to draw the same conclusion. Correlations between markers were studied as described elsewhere [15]. Briefly, the R package corrplot [16] was used to calculate Spearman’s correlation coefficients and to arrange them in the form of a correlation map. A power analysis was conducted to select significant correlations. The strength of the correlation was judged using correlation coefficients of >0.70 as strong correlations, 0.30–0.70 as moderate correlations, and <0.3 as weak correlations. A linear regression model was used to account for the variability in the treatment outcome as measured by the AHI reduction after the intervention. Nominal variables were included in the model as dummy variables. The R function lm was used. The best set of independent predictors of treatment effectiveness was selected, according to significance in the regression analysis. These results were also confirmed with a stepwise forward-backward linear regression analysis. In this process, the interaction between predictors was also carefully evaluated [17,18].

## 3. Results

In this study, we evaluated the possibility of identifying OSA patients who would benefit from MAD treatment by performing an awake UA functional endoscopy. For this purpose, during the awake UA functional endoscopy subjects were instructed to advance their mandible maximally, and they were divided into different groups according to the soft tissue response. Three types of responses were identified: enlargement (group A), stretch (group B), and unchanged (group C). The results of this test were used in combination with other noninvasive indexes to predict the treatment outcome in terms of AHI reduction. These indexes included the Epworth scale, snoring index, OAAT score, and the AHI assessment before the MAD treatment. The possible effects of age and sex were also evaluated. The main demographic and clinical characteristics of the patients are summarized in Table 1. Continuous variables are reported as the median and IQR.

In Figure 1, we compare the AHI values before and after MAD treatment for the three groups. The data lie in approximately the same range for all the groups. Scatter points are depicted in pairs, as indicated by the black dashed lines. Each pair of measurements pertains to the same patient before and after the treatment. The results of a paired-sample Wilcoxon test are reported on each plot, showing the presence of statistically significant differences in the three cases. This indicates that the treatment reduced sleep apnea in all the studied groups. However, a qualitative inspection of the data shows that a dramatic AHI reduction occurred for group A (expansion) and group B (stretch), while a modest-albeit significant-decrease occurred for group C (unchanged). The smaller *p*-values for group A and group B support this finding but also reflect a larger sample size, especially for group A.

To investigate the extent of the MAD-induced modifications more in-depth, we compared the achieved AHI perceptual reduction (Figure 2A) and the AHI reduction (Figure 2B) in the three groups.

These quantities are defined as follows:(1)AHIreduction=AHIbefore−AHIafter
and
(2)$$AHI\%reduction=\frac{\mathrm{AHIdifference}}{\mathrm{AHI}\left(\mathrm{beforeMAD}\right)}$$

The results of a Kruskal–Wallis test are superimposed on each plot, together with a post hoc analysis using the Wilcoxon unpaired two-sample test. A statistically significant difference among the three groups was found in both cases. The post hoc analysis revealed that such a difference could be ascribed to group C (unchanged), which is significantly different from groups A and B. Interestingly, we did not observe any significant difference between groups A and B, which remarkably showed a very similar AHI percentage reduction with almost overlapping IQRs. A closer inspection of Figure 2B shows that despite the large *p*-value (0.28), the AHI difference in group A has a larger range than that in group B. The absence of such an effect in the AHI percentage difference (Figure 2A) probably indicates the presence of a proportional bias associated with the initial AHI values, as will be investigated more in-depth below. Taken together, the results of Figure 1 and strongly suggest that our noninvasive advancement mandibular test could be used to predict the outcome of the intervention and the extent of the AHI reduction. To investigate whether the mandibular test could be used as an independent predictor, we compared all the indexes measured before the intervention among the three groups, which showed no significant differences (Figure 3).

For the sake of completeness, we also investigated the presence of correlations among the different markers, taking into account the determination performed before and after the intervention. For this purpose, Spearman’s correlation coefficients were computed and displayed as a correlation map (Figure 4). This map does not display the typical symmetric form, as we decided to plot all the calculated correlations below the main diagonal and only the significant correlations above the main diagonal (α = 0.05). As a general comment, all the significant values showed the presence of a moderate (ρ > 0.3) or strong (ρ > 0.7) correlation. Interestingly, no significant correlation was found between age and any of the investigated indexes.

An attempt to predict the extent of the MAD-induced modifications in terms of AHI reduction was made using a linear regression analysis. For this purpose, we first included in our model all the markers measured before the intervention as independent variables. The results of the advancement mandibular test and sex were included as dummy variables. The regression model pointed out that the only significant variables were the AHI value before the intervention and class membership in group A (expansion), B (stretch), or C (unchanged). This finding was also strengthened by a stepwise regression, which selected the same two variables (see Patients and Methods for more details).

Following this analysis, we showed the AHI reduction as a function of the initial AHI for the three groups (Figure 5A). A linear trend was fitted to the data, and the regression lines were displayed together with the corresponding 95% confidence bands. The outcomes of the regression models are summarized in Table 2.

Interestingly, intercepts are consistent with zero at the 0.05 confidence level in all cases. This might indicate that, in the absence of obstructive sleep apnea (AHI = 0), the treatment is not expected to cause any improvement, on average. The AHI reduction increased linearly with the AHI value before the intervention, indicating that a large reduction is possible only in the case of a large initial AHI. A dramatic difference between the slope trend of group C and those of groups A and B reflects a larger AHI reduction for patients belonging to the latter classes. The confidence bands in Figure 5A can be used to estimate the average AHI reduction as a function of the initial AHI together with the corresponding 95% CI. In Figure 5C,D we reported the corresponding prediction intervals. For the sake of clarity, we decided to plot them in separate figures. These bands can be used to predict the expected AHI reduction for an individual subject who has to be treated with a MAD, given his/her class membership in group A, B, or C and his/her initial AHI. As an example (red dashed lines in Figure 5B), a subject belonging to group A with an initial AHI of 60 is expected to obtain a significant AHI reduction ranging between 25 and 52 (95% CI).

## 4. Discussion

The prediction of the effectiveness of MAD in the treatment of OSA is a major problem that significantly limits the use of this therapeutic approach. Other aspects that limit the expansion of MAD use in the treatment of OSA are the difficult calibration of the device, the need for dentists specialized in sleep medicine and the high cost of the device associated with the uncertainty of the therapeutic result. In our experience, the most common concerns of patients regarding MAD therapy are related to the high cost of the device (in Italy, it is not supported by the National Health System) and to the uncertainty of therapy success. Many authors suggest diagnostic analysis or exams to predict the outcome of MAD therapy and exclude non-responder patients. However, different authors [19,20] affirmed that it is very difficult to find a reliable diagnostic predictor because there are nonanatomic pathophysiologic features, i.e., some abnormalities in ventilatory control, pharyngeal muscle responsiveness, and an arousal threshold are present in most OSA patients.

In the literature [21,22], it is well reported how DISE can provide a good indication of the chances of success with MAD therapy. For many authors [23,24], DISE is a valid diagnostic and prognostic exam, but it has many limitations:
-it is based on drug-induced sleep, and there are documented problems and concerns about the safety, efficacy, and methods of administration of these drugs;-patient hospitalization is needed for procedure execution;-it is performed during an intermittent short sleep cycle and does not permit examination of multiple airway levels simultaneously during the same cycle;-there are no clear standards for therapy with MAD planning based on data of the degree/pattern of obstruction;-the validity and reliability of DISE results are yet uncertain;-it is costly and not widely available for many patients.

To avoid the aforementioned limitations of DISE, we think that functional endoscopy of the upper airway is a simple, low-cost, convenient, informative, and noninvasive method. Many authors have tried to demonstrate the association between functional endoscopy of the upper airway and MAD therapy with few results. In the last few years, many authors have begun to demonstrate the capacity of functional endoscopy of the upper airway to discriminate responder and non-responder patients. In 2014, Sasao [2] affirmed that functional endoscopy may have significant clinical utility in predicting the success of oral appliance treatment. In 2016 and 2019, Hokuno [25,26] affirmed that functional endoscopy is an optimal predictor of responder patients with a high level of predictive accuracy (a sensitivity of 85.7%, specificity of 80.8%, and positive predictive value of 85.7%) and has negative predictive value for maximum mandibular protrusion.

Based on our experience the awake UA endoscopy is a valid prognostic exam for discriminating responder and non-responder patients. In our experience, the Muller maneuver has no predictive function and is found to be completely unreliable. Our results demonstrated that there are significant differences among the three groups in many terms. We observed a significant reduction in the AHI index in patients with a severe apnea index compared to those with a moderate or mild apnea index. Interestingly, the data show that a substantial AHI reduction occurred for group A (expansion) and group B (stretch), while a modest, albeit significant, decrease was measured for group C (unchanged). Interestingly, we did not observe any significant difference between groups A and B, which, remarkably, showed a very similar AHI percentage reduction. In our opinion, this could indicate that in many patients, the muscular contraction due to the presence of an FN in the UA hides the enlargement of the tissue so that only minimal movements and a stretch of the soft tissue is visible.

Our statistical analysis demonstrates the correlation between the pre- and post-treatment AHIs and improvements in the OAAT and Epworth Sleepiness Scale scores.Finally, our results indicate the possibility of identifying responder patients and predicting a range of posttreatment AHI index values. We believe that the main limitation of this study is the subjectivity of the analysis of the airway response to mandibular advancement. In the future, it would be interesting to consider the correlation between other assessments, like Friedman’s classification, and the response to MAD therapy [27].

## 5. Conclusions

We think that UAs awake functional endoscopy can be a discriminant exam to distinguish responder vs. non-responder patients in OSA therapy with MADs. The results obtained in this study, supported by results evidenced in the international literature, allow us to compare the predictive value of this exam with the results obtained with DISE but with minor risks, costs, and problems for the patients. Finally, the possibility of predicting the posttreatment AHI can help clinicians understand whether MAD therapy is an optimal choice for patient treatment.

## Figures and Tables

**Figure 1 ijerph-18-02393-f001:**
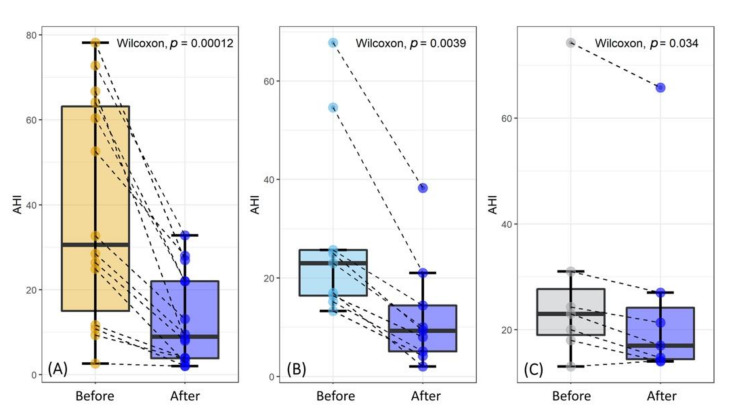
Apnea-hypopnea index (AHI) value before and after the MAD treatment for the investigated groups. Subjects were divided into groups according to the soft tissue response to the advancement mandibular test. Three types of responses were considered: expansion (**A**), tension (**B**), and unchanged (**C**).

**Figure 2 ijerph-18-02393-f002:**
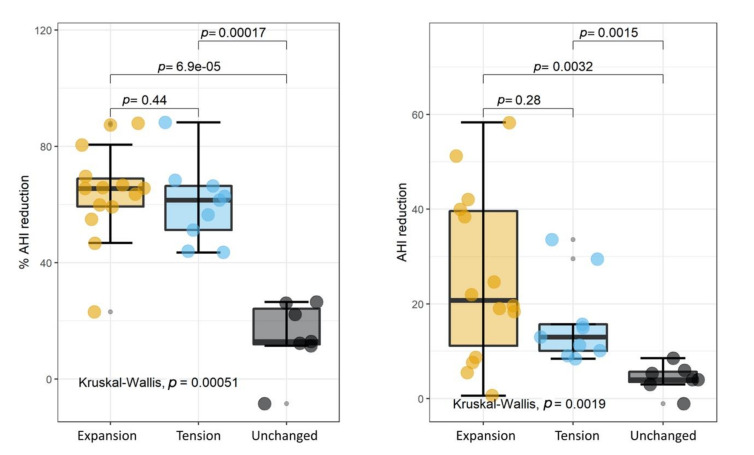
Percentage of AHI reduction (**A**) and AHI reduction (**B**) in the three investigated groups.

**Figure 3 ijerph-18-02393-f003:**
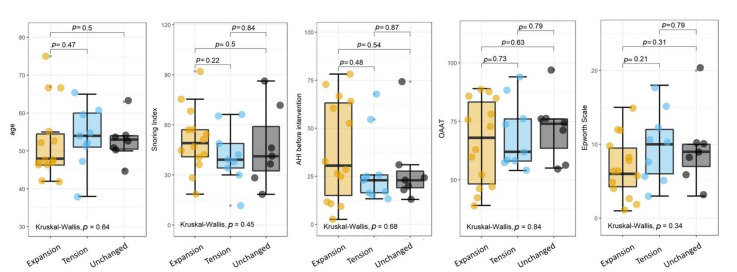
Indexes measured before the intervention in the three experimental groups.

**Figure 4 ijerph-18-02393-f004:**
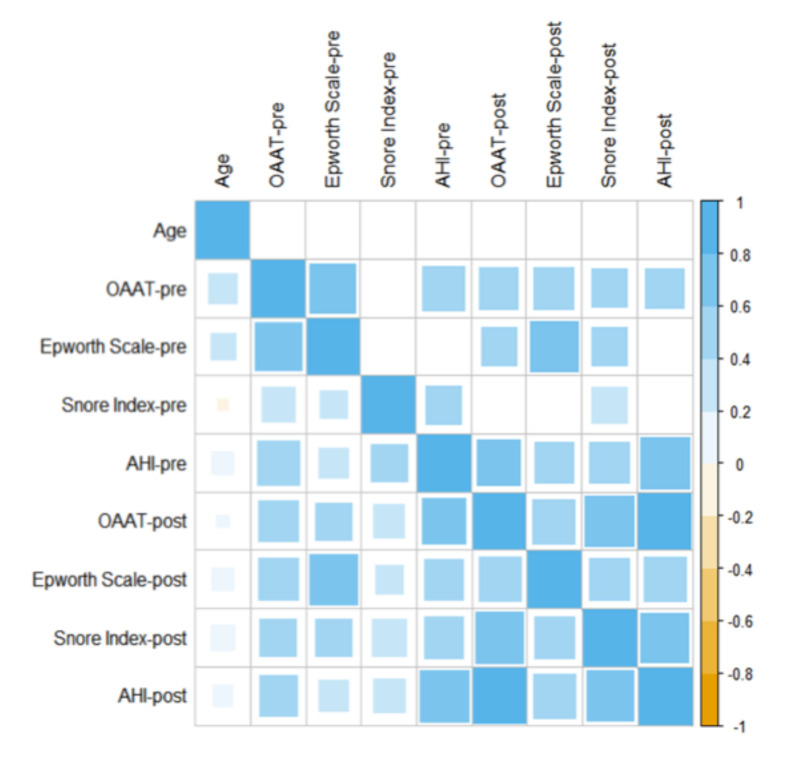
Spearmen’s correlation between different markers.

**Figure 5 ijerph-18-02393-f005:**
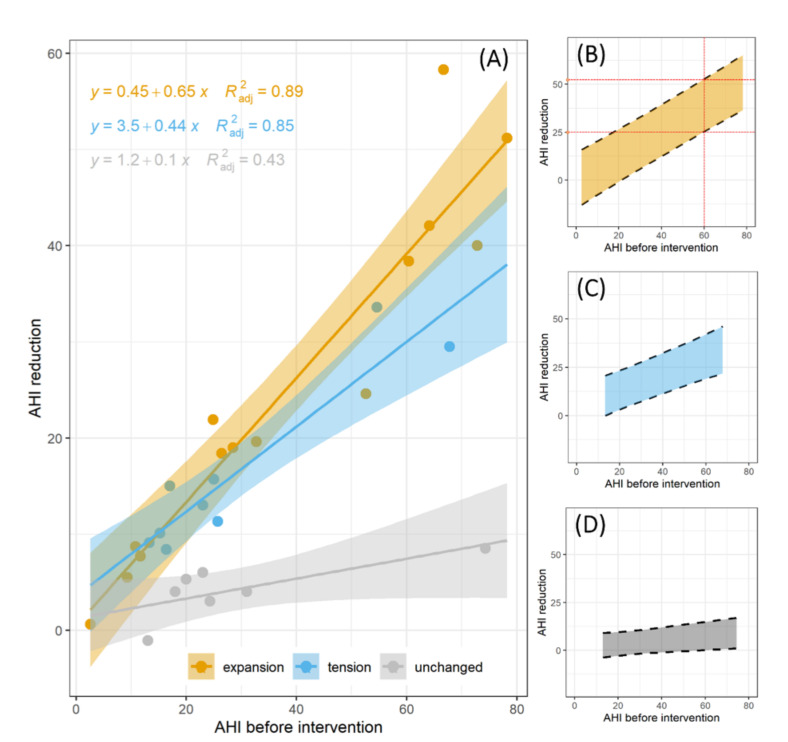
AHI reduction is a function of the initial AHI for the three groups (**A**). A linear trend was fitted to the data and the regression lines were displayed together with the corresponding 95% confidence bands. Prediction band of the linear regression model in the three groups: expansion (**B**), tension (**C**), and unchanged (**D**).

**Table 1 ijerph-18-02393-t001:** Demographic and clinical characteristics of patients, (post-treatment evaluations are performed 6 months after the start of the MAD treatment).

		All Patients	Enlargement	Stretch	Unchanged	*p*-Value
1	Number	30	14	9	7	0.273
2	Age, years [median (IQR)]	52 (8)	48 (8.25)	54 (9)	53 (3.5)	0.640
3	Male [N (%)]–Female [N (%)]	23 (77)–7 (23)	11 (79)–3 (21)	5 (56)–4 (44)	7 (100)–0 (0)	0.173
4	Pre-treatment AHI [median (IQR)]	25,35 (39.2)	42,65 (42.25)	23 (9.3)	23 (8.65)	0.680
5	Post-treatment AHI [median (IQR)]	14 (14)	8,95 (18.15)	9.65 (8775)	17 (9.75)	0.130
6	Pre-treatment Epworth Scale [median (IQR)]	8 (5.5)	6 (5.25)	10 (6)	9 (3)	0.340
7	Post-treatment Epworth Scale [median (IQR)]	5 (4)	4 (3.25)	7 (6)	5 (0.5)	0.420
8	Pre-treatment Snoring Index [median (IQR)]	43.45 (20.775)	49.15 (16.225)	39.2 (14.7)	41.3 (26.5)	0.450
9	Post-treatment Snoring Index [median (IQR)]	21.24 (24.55)	16 (12.15)	11.3 (17.8)	36.2 (25.4)	0.011
10	Pre-treatment OAAT [median (IQR)]	72 (21.25)	68 (35)	62 (18)	74 (1.,5)	0.840
11	Post-treatment OAAT [median (IQR)]	40.5 (27.5)	39 (24)	34 (19)	58 (26.5)	0.037

*p*-values refer to the comparison among the group’s enlargement, stretch, and unchanged. Different tests were used according to the type of data. Line 1, χ^2^ test for goodness of fit with an expected frequency of 0.33 for each group. Line 2: Kruskal–Wallis test. Line 3: Fisher’s Exact. Lines 4-11: Kruskal–Wallis test.

**Table 2 ijerph-18-02393-t002:** Results of the regression model. The significance of the regression parameters is indicated as a superscript.

Group	Regression Coefficient	*p*
Expansion	$\mathrm{AHI}\;\mathrm{reduction}\;=\left(-0.32\pm 2.97\right)+{\left(0.65\pm 0.06\right)}^{\;***}\times \mathrm{AHI}\;\mathrm{before}\;\mathrm{intervention}\;$	4.7 e–7
Tension	$\mathrm{AHI}\;\mathrm{reduction}\;=\left(4.51\pm 2.50\right)+{\left(0.42\pm 0.08\right)}^{\;***}\times \mathrm{AHI}\;\mathrm{before}\;\mathrm{intervention}$	0.00044
Unchanged	$\mathrm{AHI}\;\mathrm{reduction}\;=\left(1.23\pm 1.53\right)+\left(0.10\pm 0.04\right)\cdot \times \mathrm{AHI}\;\mathrm{before}\;\mathrm{intervention}$	0.06

The following convention was adopted: 0 “***”; 0.001

## Data Availability

The authors confirm that all data underlying the findings are available from the authors on request.

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
