# Peer review of "Functional Upper Airway Space Endoscopy: A Prognostic Indicator in Obstructive Sleep Apnea Treatment with Mandibular Advancement Devices"

_ijerph, 2021, doi:10.3390/ijerph18052393_

Round 1
Reviewer 1 Report
Paper well organized and statistically supported.
Otherwise, there are major revisions to discuss:
- In the introduction there no references. Please complete.
- Mandibular Advancement Device as treatment for OSA is universally accepted for mild to moderate grade of OSA, not for severe grade of the disease.
- The precise method applied for define AHI improvement is not reported: a reduction of 50% related to AHI before MAD treatment?Other? You write: "Our results demonstrated that there are significant differences among the three groups in many terms. We observed a major reduction in the AHI index in patients with a severe apnea index compared to those with a moderate or mild apnea index". What dose "major reduction in the AHI index" means?
- The awake endoscopic upper airways assessment in OSA patients must report the analysis of the relationship of base of the tongue and hypopharyngeal space. Friedman classification is an accepted method for this analysis and should be mentioned. The Authors did not mention any anatomical characteristics of base of the tongue and its relation with hypopharyngeal space.
- A subjective method reported has many limits, due to the patient ability in performing the mandibular advancement and it is stated that it is not useful assessing the maximum mandibular advancement for the MAD application, but that one sufficient to achieve a stabilization of upper airways, avoiding temporal joint functional problems.
All the above comments need to be clarified.
Author Response
Thanks for the careful analysis, below are the answers to your comments:
1.the introduction has been modified and implemented and references have been added. (page n.2 line 1,5-8-25,26-30)
2. In this study we observed a major reduction in the AHI index in patients with a severe apnea index compared to those with a moderate or mild apnea index,
"major reduction in the AHI index" means "in absolute terms" and not only in percentages, we think that the concept is shown in figure n.2 (pag 10);
the result is due to the fact that in the patients considered, the "expansion" group showed a higher AHI before intervention than the other groups.
3.Friedman Classification was not used in this study; as far as we know there are no studies that evaluate the correlation between Friedmann's classification and the response to mad. It would be interesting to consider this aspect in the future.
4.Regarding the anatomical characteristics of the base of the tongue and its relation with hypopharyngeal space, we believe that the description of the anatomical structures goes beyond our article.
5.I agree with you about the limits of a subjective method such as the one described, however, we have had good results, the article was also born with the intention of objectifying the functional upper airway space endoscopy.
The use and titration of MAD is now described more in detail.
Reviewer 2 Report
The manuscript is well written and structured. In my opinion, however, some revisions could further improve the quality:
Introduction.
- 1 line, insert reference on OSAS management and consequent MAD usage as following: Kapur VK, Auckley DH, Chowdhuri S, Kuhlmann DC, Mehra R, Ramar K, Harrod CG. Clinical practice guideline for diagnostic testing for adult obstructive sleep apnea: an American Academy of Sleep Medicine clinical practice guideline. J Clin Sleep Med. 2017;13(3):479–504.
- Line 5, cite some studies that support the concept ‘’ supported by a large international group of studies’’: Carvalho FR, Lentini-Oliveira DA, Prado LB, Prado GF, Carvalho LB. Oral appliances and functional orthopaedic appliances for obstructive sleep apnoea in children. Cochrane Database Syst Rev. 2016 Oct 5;10(10):CD005520. doi: 10.1002/14651858.CD005520.pub3. PMID: 27701747; PMCID: PMC6458031.
- ‘’ Many authors consider drug-induced sleep endoscopy (DISE)’’ report study that confirm the concept: Vicini C, Colabianchi V, Giorgio Marrano G, Barbanti F, Spedicato GA, Lombardo L, Siciliani G. Description of the relationship between NOHL classification in drug-induced sleep endoscopy and initial AHI in patients with moderate to severe OSAS, and evaluation of the results obtained with oral appliance therapy. Acta Otorhinolaryngol Ital. 2020 Feb;40(1):50-56. doi: 10.14639/0392-100X-2290. Epub 2019 Sep 30. PMID: 31570902; PMCID: PMC7147537.
- ‘’ Patients with a sufficient increase in UA width and a reduction in snoring may be valid candidates for MAD therapy. ‘’ Indicate authors that stated your affirmation and type of study, patients enrolled….
- ‘’ For these reasons, we think that DISE cannot be used in the routine diagnosis of patients who are candidates for treatment with MADs’’ the concept is incorrect. The current gold standard in the choice of the oral appliance is the disclosure at DISE of a specific pattern of collapse. Nonetheless it is true that DISE has significant costs. I recommend to formulate the concept better as follows: although the DISE remains the optimum at the moment in the indication for the oral appliance, the use of a less invasive, easy to use and inexpensive method is necessary.
Add reference: Vito A, Cammaroto G, Chong KB, Carrasco-Llatas M, Vicini C. Drug-Induced Sleep Endoscopy: Clinical Application and Surgical Outcomes. Healthcare (Basel). 2019 Aug 25;7(3):100. doi: 10.3390/healthcare7030100. PMID: 31450719; PMCID: PMC6787599.
- I suggest rephrasing the purpose of the introduction to:
- We performed an observational study in patients with respiratory sleep disorders to validate the predictive role of awake functional endoscopy in mandibular advancement treatment.
- Pretreatment evaluation: please describe ENT evalutation;
- ‘’with a significant widening of the airspace’’ define the percentage of the widening ex. 1 from 100 to 75%, 2 from 75% to 50%...
- Define position of level H: ex. Primary or secondary epiglottis..
- Define thorugh percentage the improvement of O after the advancement
- Discussion
explain different treatment approach referring in with surgical ones as expressed by the following references:
- Mantovani M, Rinaldi V, Torretta S, Carioli D, Salamanca F, Pignataro L. Barbed Roman blinds technique for the treatment of obstructive sleep apnea: how we do it? Eur Arch Otorhinolaryngol. 2016 Feb;273(2):517-23. doi: 10.1007/s00405-015-3726-2. Epub 2015 Jul 21. PMID: 26194006.
- Iannella G, Magliulo G, Di Luca M, De Vito A, Meccariello G, Cammaroto G, Pelucchi S, Bonsembiante A, Maniaci A, Vicini C. Lateral pharyngoplasty techniques for obstructive sleep apnea syndrome: a comparative experimental stress test of two different techniques. Eur Arch Otorhinolaryngol. 2020 Jun;277(6):1793-1800. doi: 10.1007/s00405-020-05883-2. Epub 2020 Mar 6. PMID: 32144568.
- Mantovani M, Carioli D, Torretta S, Rinaldi V, Ibba T, Pignataro L. Barbed snore surgery for concentric collapse at the velum: The Alianza technique. J Craniomaxillofac Surg. 2017 Nov;45(11):1794-1800. doi: 10.1016/j.jcms.2017.08.007. Epub 2017 Aug 14. PMID: 28941735.
- Di Luca M, Iannella G, Montevecchi F, Magliulo G, De Vito A, Cocuzza S, Maniaci A, Meccariello G, Cammaroto G, Sgarzani R, Ferlito S, Vicini C. Use of the transoral robotic surgery to treat patients with recurrent lingual tonsillitis. Int J Med Robot. 2020 Aug;16(4):e2106. doi: 10.1002/rcs.2106. Epub 2020 Apr 2. PMID: 32223059.
- Iannella G, Vallicelli B, Magliulo G, Cammaroto G, Meccariello G, De Vito A, Greco A, Pelucchi S, Sgarzani R, Corso RM, Napoli G, Bianchi G, Cocuzza S, Maniaci A, Vicini C. Long-Term Subjective Outcomes of Barbed Reposition Pharyngoplasty for Obstructive Sleep Apnea Syndrome Treatment. Int J Environ Res Public Health. 2020 Feb 27;17(5):1542. doi: 10.3390/ijerph17051542. PMID: 32121007; PMCID: PMC7084807.
Author Response
Thanks for the careful analysis, below are the answers to your comments:
- recommended references have been added
- the percentage of the widening is now defined ( see page n.3 Endoscopy and evaluation methods), the soft tissues of level H can respond to advancement in the same way as those of the O space.
- "ENT evaluation" has been replaced with "upper airway examination" to point out examination and assessment of the anatomy of the upper airway
- based on NOHL classification the level H represent the hypopharynx and base of the tongue area, see ref n.11
Reviewer 3 Report
Gasparini et al. used a functional endoscopy to evaluate the awake upper airway changes while the patients with obstructive sleep apnea (OAS) advanced the mandible maximally. They classified these 30 patients into three groups according to the types of anatomical change (expansion, stretch, and unchanged). They found significantly reduced apnea-hypopnea index in the expansion and stretch groups after a six-month usage of the mandibular advance device (MAD), whereas patients in the unchanged group had the worst outcome. They concluded that the awake upper airway functional endoscopy is a valid prognostic exam for discriminating responders and non-responders to the MAD. This study was interesting. However, some issues should be addressed before further consideration.
Major comments:
1. Why did your participants have a good response to the mandibular advance device (i.e. none had a worse outcome)? Did you exclude patients with obesity, obstructive tonsils, or old age?
2. Please explain why patients with an expandible upper airway also had a severe degree of OSA? Please compare more anthropometric or anatomic characteristics of them.
3. Exactly, I did not believe the patients with expandable or stretch upper airway had any different characteristics because the investigator did not objectively define the degree of airway widening.
4. Please provide a more detailed protocol about the MAD adjustment. Did the investigators adjust the MAD according to the endoscopic finding? Did the participants use the MAD with good compliance? Did they undergo OSA surgery? Did they reduce their body weight?
5. Please provide your study limitations since the participants might be highly selected. This study recruited only a few patients and lacked external validation.
Minor comments:
1. Please follow the author's guidelines. The abstract and main text should be re-edited.
2. Please cite sufficient references, esp. in the introduction, methods, and discussion.
3. What is CPAP? No reference, no explanation.
4. What is the Obstructive Airway Adult Test (OAAT) ? No reference, no explanation.
5. Please provide the P-values of Table 1.
Author Response
Thanks for the careful analysis, below are the answers to your comments:
Major comments:
1. As shown in the article our results have not always been good, as shown in the statistical analysis, the unchanged group showed no significant improvement after MAD therapy . We didn't exclude patients with obesity, obstructive tonsils, or old age. (page n.2 Inclusion and Exclusion Criteria)
2. from table 1 it can be seen that in the patients of the "enlargement" group the preoperative AHI is greater than in the remaining 2 groups, however, the other indices (Epworth Scale, Snoring Index, OAAT) have only insignificant differences.
3. the degree of airway widening is now defined, ( see page n.3 line n. 18)
4. changes made ( page n. 4 line n.7)
5. In our opinion, the two main limitations concerning the study findings are the relatively small number of patients and the subjective aspect of Functional Upper Airway Space Endoscopy.
minor comments:
1.changes made
2. changes made
3. the definition of CPAP, continuous positive airway pressure therapy, has been added.
4. the reference of OAAT is now present: " Gasparini G, Vicini C, De Benedetto M, et al. Diagnostic Accuracy of Obstructive Airway Adult Test for Diagnosis of Obstructive Sleep Apnea. Biomed Res Int. 2015;2015:915185. doi:10.1155/2015/915185 "
5. P-value has been added, (page 8 table n.1)
Round 2
Reviewer 3 Report
Thank you for performing some important revisions in your modified text. However, I still have some suggestions.
- "We performed an observational study in patients with respiratory sleep disorders to validate the predictive role of awake functional endoscopy in MAD therapy" could be modified to "We performed an observational study in patients with OSA to validate the predictive role of awake functional endoscopy in MAD therapy" because you included "obstructive" AHI > 5 in this study.
- Although the authors have been involved in the development of the OAAT test, this tool is not well recognized by the readers. Please explain this test more clearly. Items, rating, the meaning of a high score, reliability, and validity should be explained. Otherwise, please remove this test from this article since it was not related to your main outcome measures if you did not want the readers to understand its meaning.
- Did any participant undergo OSA or dental surgery? Otherwise, please do not use the terms "preoperative" and "postoperative".
- Please use "past tense" in the Material and Methods.
- Can you provide any reference that confirmed > 25% enlargement of the airway space as a significant widening?
- Please let a native English speaker carefully edit this article.
Author Response
- We modified the manuscript following your advice.
- We described better the OAAT (see page 3 2nd paragraph) for further details readers can consult the references.
- We modified pre/post-operatove with pre/post- treatment, see page 3 line 10 and table1.
- Changes made.
- No, 25% is a cut-off value that we consider to discriminate between the two groups, certainly, it’s a subjective evaluation and it is in fact the main limitation of this study, as now expressed at the end of the discussion. ( page 7).
- the English has been re-edited.